# Chikungunya Virus Asian Lineage Infection in the Amazon Region Is Maintained by Asiatic and Caribbean-Introduced Variants

**DOI:** 10.3390/v14071445

**Published:** 2022-06-30

**Authors:** Geovani de Oliveira Ribeiro, Danielle Elise Gill, Endrya do Socorro Foro Ramos, Fabiola Villanova, Edcelha Soares D’Athaide Ribeiro, Fred Julio Costa Monteiro, Vanessa S. Morais, Marlisson Octavio da S. Rego, Emerson Luiz Lima Araújo, Ramendra Pati Pandey, V. Samuel Raj, Xutao Deng, Eric Delwart, Antonio Charlys da Costa, Élcio Leal

**Affiliations:** 1Laboratório de Diversidade Viral, Instituto de Ciências Biológicas, Universidade Federal do Pará, Belem 66075-000, Brazil; geovanibiotec@gmail.com (G.d.O.R.); endrya.ramos@gmail.com (E.d.S.F.R.); fvillanova@gmail.com (F.V.); 2Instituto de Medicina Tropical da Faculdade de Medicina da Universidade de São Paulo, São Paulo 05403-000, Brazil; danielleegill@yahoo.com (D.E.G.); va.morais@usp.br (V.S.M.); charlysbr@yahoo.com.br (A.C.d.C.); 3Public Health Laboratory of Amapa-LACEN/AP, Health Surveillance Superintendence of Amapa, Macapa 68905-230, Brazil; edcelhamanu@hotmail.com (E.S.D.R.); fredjulio@gmail.com (F.J.C.M.); farmarlisson@hotmail.com (M.O.d.S.R.); 4General Coordination of Public Health, Laboratories of the Strategic Articulation, Department of the Health Surveillance Secretariat of the Ministry of Health (CGLAB/DAEVS/SVS-MS), Brasília 70719-040, Brazil; emerson.araujo@saude.gov.br; 5Centre for Drug Design Discovery and Development (C4D), SRM University, Delhi-NCR, Rajiv Gandhi Education City, Sonepat 131029, Haryana, India; ramendra.pandey@gmail.com (R.P.P.); deanacademic@srmuniversity.ac.in (V.S.R.); edelwart@vitalant.org (E.D.); 6Vitalant Research Institute, 270 Masonic Avenue, San Francisco, CA 94118, USA; xdeng@vitalant.org; 7Department Laboratory Medicine, University of California San Francisco, San Francisco, CA 94118, USA

**Keywords:** chikungunya, metagenomics, virome, amazon region

## Abstract

The simultaneous transmission of two lineages of the chikungunya virus (CHIKV) was discovered after the pathogen’s initial arrival in Brazil. In Oiapoque (Amapá state, north Brazil), the Asian lineage (CHIKV-Asian) was discovered, while in Bahia state, the East-Central-South-African lineage (CHIKV-ECSA) was discovered (northeast Brazil). Since then, the CHIKV-Asian lineage has been restricted to the Amazon region (mostly in the state of Amapá), whereas the ECSA lineage has expanded across the country. Despite the fact that the Asian lineage was already present in the Amazon region, the ECSA lineage brought from the northeast caused a large outbreak in the Amazonian state of Roraima (north Brazil) in 2017. Here, CHIKV spread in the Amazon region was studied by a Zika–Dengue–Chikungunya PCR assay in 824 serum samples collected between 2013 and 2016 from individuals with symptoms of viral infection in the Amapá state. We found 11 samples positive for CHIKV-Asian, and, from these samples, we were able to retrieve 10 full-length viral genomes. A comprehensive phylogenetic study revealed that nine CHIKV sequences came from a local transmission cluster related to Caribbean strains, whereas one sequence was related to sequences from the Philippines. These findings imply that CHIKV spread in different ways in Roraima and Amapá, despite the fact that both states had similar climatic circumstances and mosquito vector frequencies.

## 1. Introduction

The Chikungunya virus (CHIKV) is transmitted mainly by two vectors, the *Aedes aegypti* and *Aedes albopictus* mosquitoes, which are widely distributed in different continents, including America [1,2,3], favoring its dissemination into new areas and contributing to its emergence, re-emergence, and outbreaks in different parts of the world [4]. Humans, once infected, can develop the disease, the clinical manifestations of which include symptoms such as fever, headache, nausea, fatigue, myalgia, arthralgia, and rash [5]. The recovery time ranges from weeks to years. In general, clinical severity is associated with increasing age [6]. However, there are cases in which some patients (3 to 28%) do not manifest clinical symptoms [7].

CHIKV belongs to the *Togaviridae* family and is classified as an Alphavirus. The genome is made up of two open reading frames (ORFs) that encode two polyproteins that are then processed into four non-structural proteins (nsP1–nsP4) and five structural proteins (capsid, E3, E2, 6K, and E1) [8]. A phylogenetic study revealed that CHIKV is divided into four genotypes or lineages: Asian, Indian Ocean (IOL), East, Central, and South Africa (ECSA), and West Africa (WA) [9].

Originally, CHIKV was isolated in 1952 on the Makonde plateau in Tanzania [10,11]. Later, it was detected in different locations in Africa, Asia, Europe, and in the Indian and Pacific oceans [12]. In South America, CHIKV was first introduced in late 2013 from the Caribbean islands [13]. Since then, it has spread to different countries in South America. In Brazil, the first detection of autochthonous CHIKV cases was confirmed in September 2014 through two independent introductions [14]. The Asian/Caribbean lineage (CHIKV-Asian), first identified in September 2014 in the Oiapoque municipality, state of Amapá, followed by the East-Central-South African (CHIKV-ECSA) lineage, which was introduced in Brazil in the city of Feira-de-Santana, state of Bahia [14,15]. Since then, several autochthonous cases have been described [5,16]. Likewise, several imported cases were reported [17,18]. In Roraima state, in the Amazon region, CHIKV-Asian was initially introduced in 2015, and a large CHIKV outbreak occurred in 2017, caused by an ECSA-lineage [18].

The Brazil–Guyana transboundary zone (ZTBG) is in the north of the South American continent, with a hot and humid climate. The ZTBG is home to four municipalities: Saint-Georges de L’Oyapoque, Camopi, and Ouanary on the Guyanese side, and Oiapoque on the Brazilian side. The municipality of Oiapoque is located 590 kilometers from Macapá, the state capital of Amapá (Brazil). The municipality of Oiapoque, with a population of 28,000 people, is ZTBG’s largest and most important commercial center, with considerable movement of people between Brazil and Guyana.

Since the arrival of the CHIKV in South America from the Caribbean Islands, French Guyana was one of the countries most affected by the epidemic [19]; promptly, the municipality of Oiapoque also suffered from the Chikungunya epidemic (with 1541 cases between 2014 and 2016), thus demonstrating that this border region is an important gateway for the emergence of new viral epidemics. In this study, we report ten near-full length CHIKV sequences detected in individuals from Amapá, north Brazil. All of these sequences are from the CHIKV-Asian lineage, and nine of them form a single monophyletic clade, implying a local transmission cluster. One of the Amapá sequences was found to be closely related to strains from the Philippines, implying that CHIKV-Asian transmission in the Amazonian region has multiple strains originated from distinct geographical locations.

## 2. Materials and Methods

### 2.1. Sample Collection

A total number of 824 serum samples from individuals presenting with symptoms compatible with arbovirus infection were obtained from the LACEN (Laboratório Central) of the state of Amapá, of which 96, 240, 283, and 205 samples were available from the years 2013, 2014, 2015, and 2016, respectively. Additionally, epidemiological data, such as the date of symptom onset (i.e., fever with headache, myalgia, arthralgia, weakness, etc.), the date of sample collection, sex, age, and municipality of residence were collected for molecular diagnostics.

### 2.2. Sample Processing and Quantitative Real-Time RT-PCR

A Roche MagNA Pure 2.0 automatic nucleic acid extraction machine was used to extract viral RNA from the samples (MagNA Pure LC instrument, Roche Applied Science, Indianapolis, IN, USA.). The extraction reagent kits were from Roche’s MagNA Pure LC Total Nucleic Acid Isolation Kit High Performance, Version 8, and the methodology used was that stated in the kit instructions. Each sample was extracted with a volume of 200 μL of blood plasma; if the sample did not have a total volume of 200 μL, PBS was added to the sample and the contents of the sample tube were gently stirred with a pipette. Each sample had a final elution volume of 60 μL. The samples were kept in a −80 °C freezer after extraction. After that, the samples were put through a sequence of qPCR tests. To begin, all samples were subjected to the BIORAD (Bio-Rad Laboratories, Inc.; Hercules, California) ZDC (Zika, Dengue, Chikungunya-ZDC-PCR) Multiplex qPCR Assay. The assay was carried out according to the manufacturer’s protocol, which is included with the kit, using 5 μL of extracted RNA. The samples that tested positive for the ZDC assay were sent to NGS for analysis.

### 2.3. Library Preparation and Next-Generation Sequencing

The preparation of the next-generation sequencing library was performed as described by da Costa [20]. To eliminate host and bacterial cellular debris, 0.3 mL of the sample was centrifuged at 12,000× *g* for 5 min at 8 °C and the supernatant was filtered through a 0.45 M Millipore filter (Billerica, MA, USA). The filtrate was then treated for 1.5 h at 37 °C with a mixture of nucleases to decrease nucleic acids, keeping only the infectious viral nucleic acids, which are protected from digestion by their capsids. The ZR & ZR-96 DNA/RNA kits were then used to extract total nucleic acid (Zymo Research, Irvine, CA, USA). The elution volume was 0.05 mL of nuclease-free water, according to the manufacturer’s guidelines. The SuperScript III kit (Life Technologies, Grand Island, NY, USA) was used to make the first strand of cDNA, and the Klenow FRAGMENT kit was used to make the second strand (New England Biolabs, Ipswich, MA, USA). After that, the final product was sent to Nextera XT (Illumina, San Diego, CA, USA) for library preparation. Illumina software was used to demultiplex the paired-end, 300 pb sequences generated by MiSeq. The data was then run via the Blood Systems Research Institute’s “virus discovery” pipeline on supercomputers (Deng et al., 2015). Bowtie2 was used to filter the sequences, excluding human, bacterial, and fungal sequences. SOAPdenovo2, Abyss, meta-Velvet, CAP3, Mira, and SPADES algorithms were later used to reconstruct viral genomes. BLASTx and BLASTn were used to analyze the contigs. Using Geneious R8, the quality and coverage of the entire or partial genomes were assessed (Biomatters, San Francisco, CA, USA).

### 2.4. Phylogenetic and Bayesian Analysis

Firstly, we submitted the sequences generated in this study to a genotyping analysis using the phylogenetic arbovirus subtyping tool, available at http://genomedetective.com/app/typingtool/chikungunya (accessed on 8 May 2022) [21]. To investigate the phylodynamics of CHIKV in the state of Amapá, we downloaded all sequences assigned as CHIKV from GenBank (*n* = 6232), submitted them to Genome detective, and selected only Asian and Caribbean genotype sequences out of the remaining 1034 sequences. This dataset, plus Brazilian sequences, was aligned using MAFFT [22] and edited using AliView [23]. Partial, poorly aligned, and identical sequences were removed from the dataset. A final dataset, with 257 sequences, was used for phylogenetic analysis. We estimated maximum-likelihood phylogenies in PhyML [24] by using the best-fit model of nucleotide substitution, as indicated by the jModelTest application [25]. To investigate the temporal signal in our CHIKV dataset, we regressed root-to-tip genetic distances from this ML tree against sample collection dates using TempEst v 1.5.1. Time-scaled phylogenetic trees were inferred by using the BEAST package v.1.10.4 [26]. We employed a model selection analysis using both path-sampling and stepping stone models to estimate the most appropriate model combination for Bayesian phylogenetic analysis, and the best fitting model was the TN-93 plus Gamma correction substitution model with a Bayesian skyline coalescent model. Phylogeographic analyses were applied as an asymmetric model of location transitioning coupled with the Bayesian stochastic search variable selection (BSSVS) procedure. We complemented this analysis with Markov jump estimation that counts location transitions per unit time along the tree. The Monte Carlo Markov chains ran long enough to ensure stationarity and adequate effective sample size (ESS) of >200. A final maximum clade credibility tree was generated by summarizing the results of Bayesian phylogenetic inference and viewed in FigTree software [20,27].

### 2.5. Epidemiological Data Compilation

Data of weekly notified CHIKV cases in Brazil and the state of Amapá were collected from Sistema de Informação de agravos de notificação-SINAN (http://portalsinan.saude.gov.br/sinan-dengue-chikungunya (accessed on 24 April 2022)) and bulletins from the Superintendence of Health Surveillance of State of Amapá (https://svs.portal.ap.gov.br/publicacoes (accessed on 24 April 2022)).

## 3. Results

### 3.1. PCR Assay

We processed 824 samples using the ZDC-PCR assay, of which 788 were negative and 36 samples (4.3%) were positive (0 from 2013, 8 from 2014, 23 from 2015, and 5 from 2016). Of the 36 positive samples, 24 were DENV (5 from 2014; 16 from 2015; and 3 from 2016), 11 were CHIKV (3 from 2014; 7 from 2015; and 1 from 2016), and 1 was ZIKV (from 2016). Partial results of this serum survey were previously published [20,27]. Regarding the 11 CHIKV positive samples, the average RT-PCR cycle threshold was 28.02 (ranging from 20.2 to 37.35), and was from patients with an average age of 32 years of age, of which the majority (67.6%) were female (Table 1).

### 3.2. Location of Sample Collection

The patients lived in the cities of Macapá (capital of the state of Amapá), Laranjal do Jari, and Porto Grande Municipalities (Figure 1). At the time of sample collection, patients had the following symptoms: fever in the last seven days, pain in the muscle, localized exanthema, and cough.

### 3.3. Next Generation Sequencing and Genotyping Tree

From the CHIKV samples, we were able to generate 10 complete or near-complete genome sequences and deposited them in GenBank at NCBI access numbers OL343608–OL343617. The identification of CHIKV genotypes was performed using phylogenetic analysis of full-length genome datasets and using an online tool (http://genomedetective.com/app/typingtool/chikungunya (accessed on 8 May 2022)). We also inferred a maximum likelihood tree that includes more CHIKV-Asian references and variants from other countries to give more support to the classification of our sequences (tree not shown). Both approaches indicated that all sequences generated in this study belong to the Asian lineage (Figure 2).

### 3.4. Phylogenetic Analysis

We applied the maximum likelihood (ML) criterion to construct phylogenetic trees of complete genomes of the CHIKV-Asian lineage using sequences from South Asia, the Caribbean, and the American continent. The aim here is to understand the relatedness of Brazilian sequences to CHIKV-Asian lineages. Initially, we used 257 CHIKV-Asian lineage sequences to construct a phylogenetic tree (Appendix A). This ML tree indicates that all sequences from the Caribbean and Americas are in a monophyletic clade previously designated the Caribbean lineage. The Caribbean lineage is related to Asian sequences, particularly strains from the Philippines detected in 2014 and 2016 (GenBank accession numbers MF773563 and MF773564, respectively). This lineage also includes one sequence from the 2015 strain from French Polynesia (KR559473) [28]. In addition, all Brazilian sequences are in the Caribbean clade, with the exception of one Brazilian sequence (695_Amapa_Brazil_2016) which is related to the sequence MF773564 from the Philippines. The clade formed by the sequences and MF773564 is at the base of the Caribbean lineage. For illustrative purposes, we also constructed a small version of the ML tree with 99 sequences (Figure 3), including all sequences from South Asian/Oceania (indicated in red in the tree), all Brazilian sequences (blue in the tree) and some Caribbean references (indicated in green in the tree). This tree equally shows the monophyletic pattern of Brazilian sequences of the Caribbean lineage that are grouped into a single clade with high statistical support (0.99). In addition, the tree indicates that the sequence 695_Amapa_Brazil_2016 is not in the Caribbean clade, and is related to one sequence from the Philippines in 2014.

### 3.5. Polyphyletic versus Monophyletic Pattern of CHIKV from Amapá

We applied a maximum likelihood hypothesis test (Shimodaira–Kishino test) to evaluate the monophyletic versus the polyphyletic pattern of our CHIKV sequences from Amapá because sequences of the Caribbean lineage have reduced diversity (less than 1% of nucleotide divergence). This low divergence, besides producing near zero branch lengths, can also impact on the grouping pattern of ML trees. To provide more support to the circulation of distinct CHIKV variants, we constructed one tree using a coalescent approach assuming that all sequences from Amapá were monophyletic, including the sequence 695_Amapa, (Figure 4a), and tested against a coalescent tree in which the sequences 695_Amapa in not monophyletic (Figure 4b). The Shimodaira–Kishino test indicated that the polyphyletic tree (Figure 4b) is the most likely tree because it has the best-log likelihood compared with the alternative monophyletic tree (i.e., 21,146.93 and 21,121.1, respectively). These results indicate that the 695_Amapa is not closely related to sequences of the Caribbean lineage.

### 3.6. Time-Scaled Tree

We applied a Bayesian coalescent approach to better understand the temporal pattern and the topology of trees of the CHIKV-Asian lineage. Initially, the linear regression of root-to-tip genetic distance against the sampling date from our dataset revealed a sufficient temporal signal (r2 = 0.79, Appendix A). Next, we used a constant population size coalescent model to infer a tree in which no CHIKV of the Caribbean-lineage sequences were included besides the Brazilian sequences (Figure 5). The estimated substitution rate of the CHIKV-Asian lineage is evolving at 6.7 × 10^−4^ substitutions per site per year. This time-scaled Maximum clade credibility tree (MCC tree) shows that nearly all Brazilian CHIKV sequences grouped in a single clade highly the posterior probability (PP = 0.99). The exception was the sequence 695_Amapa which was grouped with sequences from the Philippines. We also used a model (Skygrid) to evaluate the fluctuation of the effective population size over time. These results indicate that CHIKV was introduced in the Caribbean region in early 2013 from South Asian variants (probably by Philippian variants) (PP= 1.0). Following this introduction, the number of infections increased steadily (Appendix A).

The MCC tree also showed that the Amapá sequences were grouped into different clades, suggesting the introduction of this genotype in Amapá occurred on at least two occasions in the middle of 2014. In addition, the tree indicates that the sequence 695_Amapa is at the base of the Caribbean lineage (Figure 5).

The presence of non-synonymous substitutions already described as potential related to vector adaptability was investigated among sequences from Amapá. We observed that all 10 Amapá sequences do not carry either the substitutions A226V (E1 protein) or the L210Q (E2 protein) associated with increased CHIKV transmission in *A. albopictus* mosquitos. On the other hand, other substitutions related to the vector adaptability, such as T98A, A377V, M407L (E1 protein), G60D, and A103T (E2 protein), were present, as shown in Table 2.

## 4. Discussion

The virus has spread across the region due to the widespread movement of individuals across the Brazil–Guyana border, notably for commercial reasons, as well as the region’s vulnerable health system. Unauthorized logging and mining in the Oiapoque River basin have expanded human activity in the forest region, leading to the establishment of new communities near the municipality of Oiapoque. Brazilian miners’ ventures deep into the forest have polluted the ecosystem with heavy metals used in gold mining, resulting in increased migratory movements between neighboring countries [29].

This uncontrolled flow of people creates a unique set of challenges for local health services since these highly mobile individuals are hard to track, making it difficult to assess resource needs and plan measures at the local level [30,31]. DENV-4 [32], DENV-1 [27], and CHIKV [14] are only a few examples of viral variations that have recently been found in the Brazilian Amazon.

To better understand CHIKV evolution in the state of Amapá, we sequenced ten full-length genomes and performed a phylogenetic analysis. Almost all CHIKV sequences from Macapá and Laranjal do Jari municipalities clustered in a monophyletic phylogroup, with one sequence from Macapá municipality (695 Amapá) showing a unique clustering pattern linked to a sequence reported in the Philippines in 2014 (Genbank ID: MF773563) [28].

Furthermore, the 695 Amapá and MF773563 sequence clusters are found near the base of the Caribbean lineage. With approximately 16,000 people infected between 2014 and 2015, French Guyana became the first South American country to declare CHIKV infections autochthonous in 2014. It has been claimed that South Pacific strains gave rise to the Caribbean lineage [33,34]. More recently, it has been proposed that the Caribbean lineage was imported either directly or indirectly from Southeast Asia [28]. This possibility is strengthened by our phylogenetic research. Furthermore, we discovered one Amapá sequence that, along with the Philippian strains, forms the foundation of the Caribbean lineage’s phylloclade.

It’s worth noting that, when compared to Asian/Oceanic comparisons, all Caribbean sequences include two conventional substitutions: V226A in the E2 gene and L20M in the 6K gene. Some alterations have been shown to have an effect on CHIKV fitness, mostly by interfering with the virus’s ability to transmit. In the Indian Ocean outbreak, for example, the CHIKV E1 glycoprotein substitution A226V has been linked to increased transmission by Aedes albopictus, which is thought to have the lowest vector capacity for CHIKV transmissibility than *Aedes aegypti* [35].

This replacement has also been implicated in European outbreaks involving *Aedes albopictus* [36]. L210Q arose in IOL strains in India and was linked to greater CHIKV transmission by the *Aedes albopictus* vector [37], while V226A boosted viral dissemination in *Aedes aegypti* but not in *Aedes albopictus* [38,39].

To our knowledge, none of the Brazilian strains (Asian or ECSA lineages) contain the substitutions A226V in the E1 glycoprotein or L210Q in the E2 glycoprotein. On the other hand, the Amapá sequences, as well as other Asian strains, have threonine (T) in position 98 of the E1 glycoprotein, which limits their infectivity in *Ae. albopictus* through the substitution A226V [40]. In particular, the sequence 695_Amapa has the residues V226 in the E2 gene and L20 in the 6K gene, likewise the sequences from South Asian/Oceania. Moreover, we found six amino acid changes in Amapá sequences (nsP1, capsid, E1, and E2 proteins), none of which have been described previously. The capsid, E2, and E1 glycoproteins are responsible for recognizing the host-receptor and entry into the cell [41,42], and they are widely used for vaccine development and serodiagnostic assays [43]. These substitutions in the E1 and E2 proteins could cause viral evasion and potentially allow host-switching of the CHIKV-Asian lineage in the Amazon region [44].

On the American continent, the CHIKV-Asian lineage is widely distributed, while the CHIKV-ECSA lineage is restricted only to Brazil, Paraguay, and Haiti [45,46]. In contrast, in Brazil, the CHIKV-Asian lineage was limited to a small number of cases and was geographically restricted (predominantly in the state of Amapá), whereas the ECSA lineage was widespread [20,47]. It has been shown that CHIKV-Asian is less capable of accumulating mutations that facilitate their transmission through vectors when compared to the ECSA lineage [48]. In Roraima, a state in the north of Brazil that borders Venezuela, the CHIKV-Asian lineage was introduced in 2015 from northeastern Brazil, and a large outbreak in 2017 was caused by the ECSA genotype that gradually replaced the original CHIKV strains [18]. Although Roraima and Amapá have very similar climate conditions, the CHIKV-Asian lineage continued to be predominant in the state of Amapá. It is important to mention that *Ae. aegypti* and *Ae. albopictus* are endemic in the Amazon region [49]. In addition, the dissemination of CHIKV-Asian by *Ae. albopictus* might be restricted due to the genetic background of the virus [39]. This pattern of CHIKV strain replacements in the Amazon region is probably affected by multiple factors such as the ecological community, human behavior, and the genetics of the virus.

The modest number of CHIKV sequences we were able to recover from the samples was perhaps the most significant limitation of our research. This has restricted our ability to examine the genetic diversity of CHIKV in depth. Multidisciplinary research will be required to understand the key factors that contribute to the transmission dynamics of CHIKV in the Amazon.

## 5. Conclusions

Despite the low frequency of CHIKV in our samples, we observed various strains circulating in the Amapá, indicating that CHIKV-Asian transmission in the Amazonian region originated from different geographical locations.

## Figures and Tables

**Figure 1 viruses-14-01445-f001:**
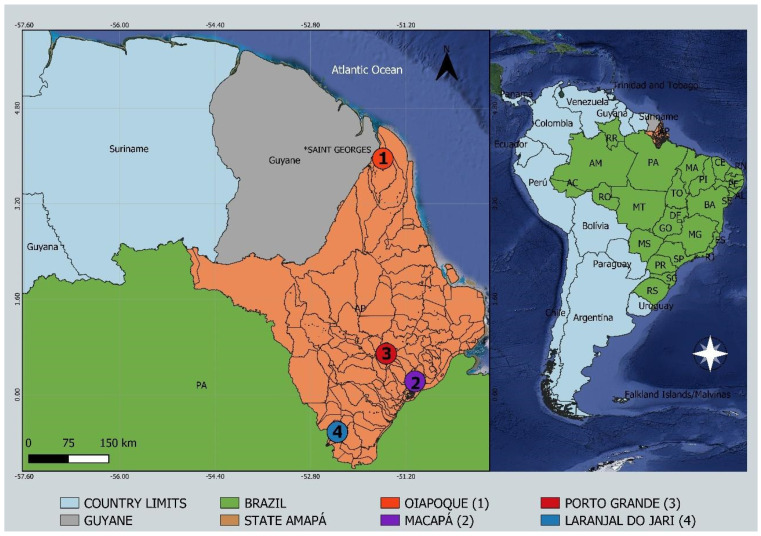
Study sites in the state of Amapá, northern Brazil, showing locations of sequenced cases of CHIKV-Asian outbreak. Circles indicate municipalities: (1) Oiapoque; (2) Macapá; (3) Porto Grande; (4) Laranjal do Jari.

**Figure 2 viruses-14-01445-f002:**
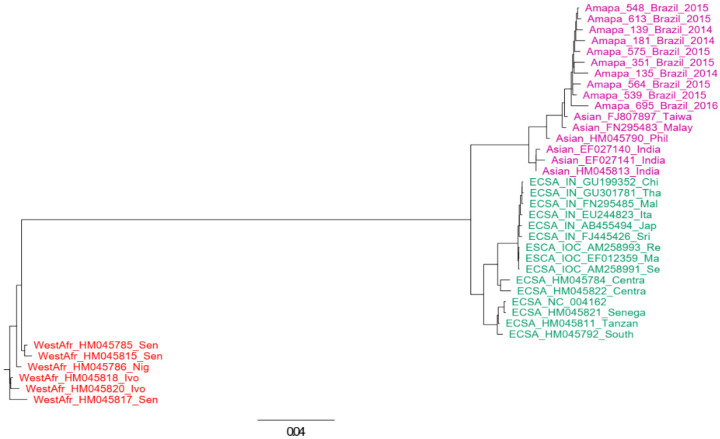
Genotyping tree of CHIKV. The tree was constructed using complete genomes of CHIKV. The maximum likelihood approach was used and the model GTR was assumed in this inference. Sequences of Asian lineage are indicated in magenta, those of West African lineage are in red, and those of ECSA are indicated in green. The horizontal bar indicates the nucleotide substitution per base.

**Figure 3 viruses-14-01445-f003:**
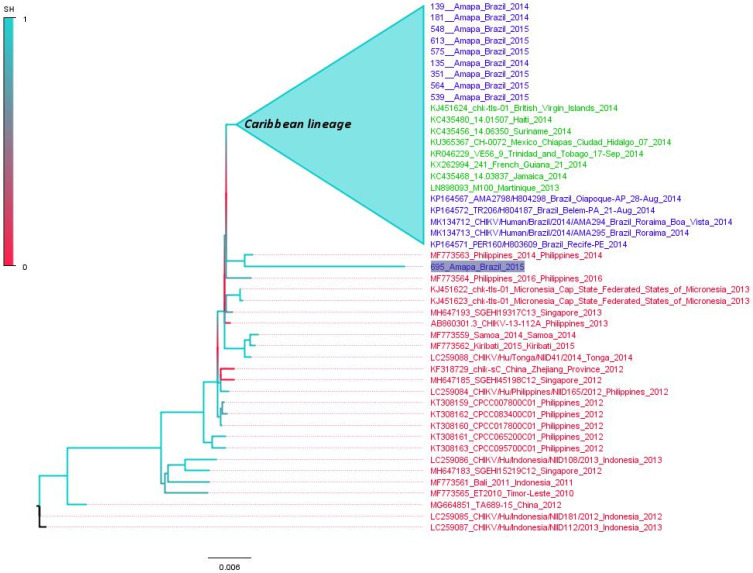
Maximum likelihood tree of CHIKV-Asian lineage. The unrooted tree was constructed using reference genomes of CHIKV-Asian lineage. Sequences from Brazil are indicated in blue. Sequences from Caribbean countries are in green color and sequences from South Asia/Oceania are in red color. The cluster composed by Caribbean sequences plus almost all Brazilian sequences is labeled. The Brazilian sequence related with a Philippine sequence is highlighted. The branch support is indicated by a color scale of 0 to 1, and is based on the Shimodaira–Hasegawa-like test. The tree was inferred using the TN-93 model plus gamma correction. Horizontal bar indicates the nucleotide substitution per base. For better visualization of the tree some sequences were collapsed (blue triangles).

**Figure 4 viruses-14-01445-f004:**
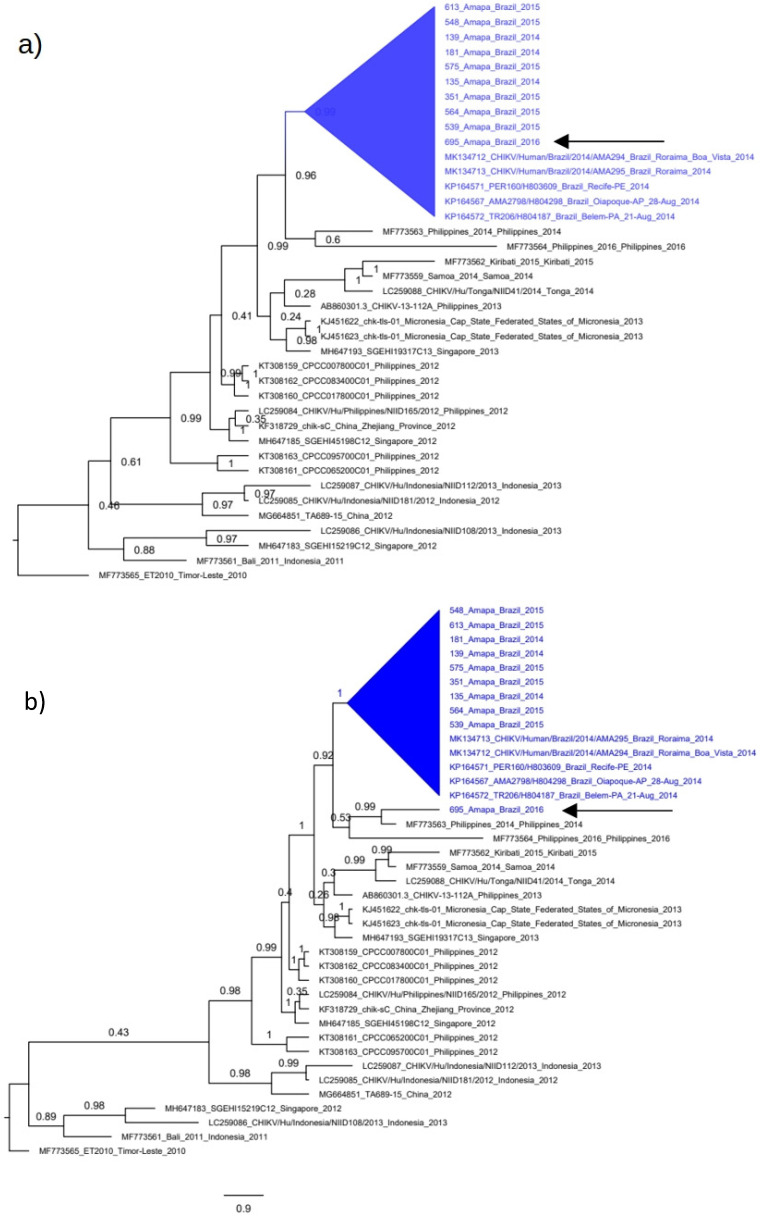
Monophyletic versus polyphyletic tree of CHIKV-Asian in Amapá. The coalescent trees were constructed using the complete genome of Brazilian CHIKV plus South Asian/Oceania references. The Bayesian approach was used and trees were constructed assuming the constant coalescent model plus the TN-93 evolutionary nucleotide model. The horizontal scale bar indicates nucleotide substitutions per site. Monophyletic clades are shown in the blue triangles in both trees. (**a**) Coalescent tree constructed assuming that the sequence 695_Amapa (indicated by arrow) is monophyletic. (**b**) Coalescent tree constructed assuming that the sequence 695_Amapa (indicated by arrow) is not within the monophyletic formed by the Brazilian sequences.

**Figure 5 viruses-14-01445-f005:**
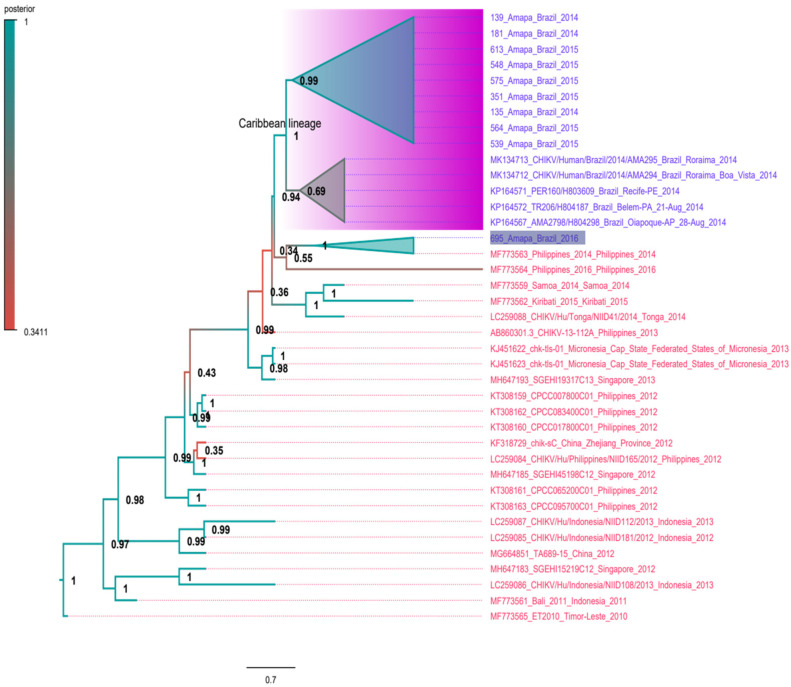
Time-scaled tree of CHIKV-Asian. The tree was constructed using complete genomes of CHIKV. The coalescent constant population size approach was used and the model TN-93 was assumed in this inference. Sequences of Asian/Oceania are indicated in red color and Brazilian sequences are in blue color. The Caribbean lineage is also indicated in the magenta rectangle. The sequence 695_Amapá is highlighted. Numbers in the nodes indicate the posterior probability. The horizontal bar indicates the nucleotide substitution per base.

**Table 1 viruses-14-01445-t001:** Epidemiological data for the CHIKV samples.

Sample ID	Collection Date (Y-M-D)	Age ^1^	Sex	Municipalities	Cycle Threshold
135	2014-10-10	54	F	Macapá	28.21
139	2014-10-24	20	F	Macapá	22.56
181	2014-10-28	29	F	Macapá	26.93
351	2015-01-09	21	M	Macapá	32.77
539	2015-02-03	16	F	Laranjal do Jari	35.61
548	2015-02-20	42	M	Macapá	20.64
564	2015-02-03	61	F	Macapá	37.35
574 *	2015-01-29	30	M	Porto Grande	NA
575	2015-02-24	30	F	Macapá	27.93
613	2015-02-25	26	F	Macapá	20.2
695	2016-03-11	28	M	Macapá	NA

NA = Not Available; F = Female; M = Male. * From this sample we were not able to sequence the CHIKV genome; ^1^ Age at collection date.

**Table 2 viruses-14-01445-t002:** Non-synonymous substitutions observed among the 10 CHIKV genomes sequenced in this study.

Gene	Substitution	Amapá Sequences	Function
E1	L19M	Yes	Unknown
	T98A	Yes	Enhanced vector adaptability of A226V
	A226V	No	Increased infectivity, transmission, and dissemination in *A. Albopictus*
E2	L210Q	No	Enhanced disseminated infection in *A. albopictus* and fitness increment of A226V variant
	V226	Yes *	Unknown
	V367A	Yes	Unknown
	V384M	Yes	Unknown
nsP1	R364K	Yes	Unknown
Capsid	V54A	Yes	Unknown
6k	L20	Yes *	Unknown

* Residues shared between 695_Amapa and sequences from South Asian/Oceania.

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
