# Peer review of "Chikungunya Virus Asian Lineage Infection in the Amazon Region Is Maintained by Asiatic and Caribbean-Introduced Variants"

_viruses, 2022, doi:10.3390/v14071445_

Round 1

Reviewer 1 Report

General Comments

This article, “Chikungunya virus Asian lineage infection in the Amazon region is maintained by Asiatic and Caribbean-introduced variants”, provides important information of the lineage and source of circulating genotypes of CHIKV in the Amazon region of Brazil. Authors have pointed out the limitation of the sample site in the broader shceme of things but as rightly put, the imitations do not deviate from the importance of their findings. However, below are a few comments that need addressing by authors before.

Method

Sample collection

Authors should rephrase the paragraph for clarity.

·         The paragraph as written suggests the numbers collected each year was a subset of the total presenting cases that year. If this is the case, what influenced the author’s decision to exclude some patients presenting arbovirus infection-like symptoms?

·         How can sample collection be arbitrarily done if authors had a strict exclusion/inclusion criteria

·         Did authors collect information on the travel history of patients?

Results

The following information was not present in any of the two previous publications from this study and should be provided in this one if available

·         Was the patient infected with Zika infected with any other arbovirus?

·         What was the percentage of patients suffering from multiple infections?

·         Was there any observable or significant correlation between multiple infection and Ct value of CHIKV positive patients

·         Was there any indication of importation of 695_Amapa into the region?

Author Response

Reviewer 1:

This article, “Chikungunya virus Asian lineage infection in the Amazon region is maintained by Asiatic and Caribbean-introduced variants”, provides important information of the lineage and source of circulating genotypes of CHIKV in the Amazon region of Brazil. Authors have pointed out the limitation of the sample site in the broader shceme of things but as rightly put, the imitations do not deviate from the importance of their findings. However, below are a few comments that need addressing by authors before.

 Method

Sample collection

Authors should rephrase the paragraph for clarity.

·         The paragraph as written suggests the numbers collected each year was a subset of the total presenting cases that year. If this is the case, what influenced the author’s decision to exclude some patients presenting arbovirus infection-like symptoms?

Resp: We changed this sentence in the new version of the manuscript. The only criteria used to select these samples were for arbovirus symptoms. The availability of samples resulted in an unequal number of samples being analyzed each year.

        How can sample collection be arbitrarily done if authors had a strict exclusion/inclusion criteria

Resp: Samples were selected based on the patient’s symptoms related to viral infection.

·         Did authors collect information on the travel history of patients?

Resp: We don't have any information on the patients' travel history; nonetheless, all of the patients were local residents from underprivileged communities with limited transportation options.

Results

The following information was not present in any of the two previous publications from this study and should be provided in this one if available

·         Was the patient infected with Zika infected with any other arbovirus?

Resp: No, this individual was only infected with the Zika virus, which was confirmed by both serological and molecular assays.

·         What was the percentage of patients suffering from multiple infections?

Resp: None of the 824 samples tested by ZDC-PCR were found to be infected by two or more viruses.

·         Was there any observable or significant correlation between multiple infection and Ct value of CHIKV positive patients

Resp: We didn't find any infections that were multiple.

·         Was there any indication of importation of 695_Amapa into the region?

Resp: This is a limitation of our sample collection, the lack of travel records of these patients. We know that all individuals are local residents with restricted conditions of travel. In particular, this individual (sample 695) is a resident of Macapa City. In the introduction, we describe the importance of Macapa City as a commercial hub connecting the North region of Brazil and countries like French Guiana and Guyana. Besides, the region also has a busy port, in the neighboring city of Santana, that concentrates the fluvial transportation coming from the interior of the state and the international transportation to and from the USA and the Caribbean region. So, Macapa is a melting pot where people and insect vectors from different regions are in close contact. Consequently, the most parsimonious explanation is that patient 695 was infected by a CHIKV lineage coming from abroad, probably on a ship from the Asian region.

Reviewer 2 Report

This manuscript by Geovani de Oliveira Ribeiro et al. describes the circulating CHIKV lineages in the Amapá state of Brazil during 2013-2016.  The authors identified by PCR 11 CHIKV positive serum samples out of 824 samples.  They recovered 10 full-length viral genomes and performed comprehensive phylogenetic analyses. The results revealed that 9 CHIKV sequences came from a local transmission cluster related with Caribbean strains, whereas only one sequence was related to Philippines strains.  

The CHIKV samples are from a 2013-2016 surveillance, not representing currently circulating strains.  Please discuss what new information can be learned from this study when compared to similar publications on CHIKV strains/lineages in Brazil.  

Please list notable mutations (such as A226V in the discussion) and their putative association with CHIKV pathogenesis, transmission, adaptability to vectors, etc. in a table. This will help readers quickly grab the major differences between the Amapa strains and Asian/Caribbean lineage.

Can the authors speculate why the Asian lineage is restricted to the Amazon region (mostly in the state of Amapá), whereas the ECSA lineage is dominant in Roraima, despite both states have similar climatic conditions and mosquito populations?

Author Response

Reviewer 2:

This manuscript by Geovani de Oliveira Ribeiro et al. describes the circulating CHIKV lineages in the Amapá state of Brazil during 2013-2016.  The authors identified by PCR 11 CHIKV positive serum samples out of 824 samples.  They recovered 10 full-length viral genomes and performed comprehensive phylogenetic analyses. The results revealed that 9 CHIKV sequences came from a local transmission cluster related with Caribbean strains, whereas only one sequence was related to Philippines strains.  

The CHIKV samples are from a 2013-2016 surveillance, not representing currently circulating strains.  Please discuss what new information can be learned from this study when compared to similar publications on CHIKV strains/lineages in Brazil.  

Resp: The Amapá state is the only place in Brazil where local transmission of Chikungunya, caused by the Caribbean/Asian lineage, has been reported. Although there were more than 1,500 cases between 2014 and 2016, only one complete genome from that outbreak, which occurred in Amapá state, is available. In this study, we found one sequence related to CHIKV strains from the Philippines. Up until now, all sequences detected in Amapá were related to CHIKV strains from the Caribbean region. Our study shows that CHIKV infection in Amapá is comprised of strains introduced from the Caribbean and also from South Asia.

Please list notable mutations (such as A226V in the discussion) and their putative association with CHIKV pathogenesis, transmission, adaptability to vectors, etc. in a table. This will help readers quickly grab the major differences between the Amapa strains and Asian/Caribbean lineage.

Resp: The main amino acid variations and their activities in the CHIKV life cycle are summarised in a table (table 2).

Can the authors speculate why the Asian lineage is restricted to the Amazon region (mostly in the state of Amapá), whereas the ECSA lineage is dominant in Roraima, despite both states have similar climatic conditions and mosquito populations?

Resp: It is, indeed, a fascinating fact. In August 2014, two sequences of the Caribbean/Asian lineage imported from Venezuela were identified in Roraima, and in July 2016, all sequences detected were of the ECSA lineage (this lineage was first detected in the state of Bahia).

Despite the fact that both Roraima and Amapá are located in the Amazon basin, there are some socioeconomic distinctions that may have an impact on CHIKV infection dynamics. Amapá is a commercial center; the state is crisscrossed by rivers, and fluvial transportation serves as a vital link between the interior (forest and smaller cities) and the capital (Macapá). Wood, ores, peach palm, and açay fruit are among Amapá's exports. A port in the neighboring city of Santana connects Amapá to Brazil's northern regions and the Caribbean countries. The Macapá airport is a well-known route for people traveling to or from Europe via Suriname.

Roraima, on the other hand, is the least populous and has the country's lowest economy. Agriculture is the backbone of the economy. There is also small-scale trade with Venezuela, with wood being the primary export. The availability of river transportation is restricted, and there is only one road linking Roraima with the Amazonas state. Flights to Belém, Manaus, and Brasilia are available from the capital city's airport (Rio Branco).

These differences in people and trade movement between Amapá and Roraima may have contributed to the spread of various CHIKV strains and the establishment of local infection by distinct lineages.

Round 2

Reviewer 2 Report

The authors have addressed my critiques.